American Society for Microbiology | Microbiology Spectrum

# Grass supplementation to a pellet-based diet fails to enrich gut microbiomes with wild-like functions in captive-bred hares

Ostaizka Aizpurua,[1] Garazi Martin-Bideguren,[1] Nanna Gaun,[1] Antton Alberdi[1]

**ABSTRACT**   Reintroducing captive-bred animals into the wild often faces limited success, with the underlying causes frequently unclear. One emerging hypothesis is that maladapted gut microbiota may play a significant role in these challenges. To investigate this possibility, we employed genome-resolved metagenomics to analyze the taxonomic and functional differences in the gut microbiota of 45 wild and captive European hares (*Lepus europaeus*), as well as to assess the impact of fresh grass supplementation to a pellet-based diet aimed at pre-adapting captive hares to wild conditions. Our analyses recovered 860 metagenome-assembled genomes, with 87% of them representing novel species. We found significant taxonomic and functional differences between the gut microbiota of wild and captive hares, notably the absence of Spirochaetota in captive animals and differences in amino acid and sugar degradation capacities. While grass supplementation induced some minor changes in the gut microbiota, it did not lead to statistically significant shifts toward a more wild-like microbial community. The increased capacity for degrading amino acids and specific sugars observed in wild hares suggests that, instead of bulk grass, dietary interventions tailored to their specific dietary preferences might be necessary for pre-adapting hare gut microbiota to wild conditions.

**IMPORTANCE** This study sheds light on the role of gut microbiota in the success of reintroducing captive-bred animals into the wild. By comparing the collection of 860 near-complete genomes of wild and captive European hares, we identified significant taxonomic and functional differences, including the absence of key microbial groups in captive hares. Grass supplementation to a pellet-based diet yielded limited success in restoring a microbiota similar to that of wild counterparts, highlighting the need for more tailored approaches to mimic natural diets. With 87% of recovered microbial genomes representing novel species, this research also enriches our understanding of microbial diversity in wildlife. These findings emphasize that maladapted gut microbiota may hinder the survival and adaptation of reintroduced animals, suggesting that microbiome-targeted strategies could improve conservation efforts and the success of animal rewilding programs.

**KEYWORDS**   reintroduction program, microbiota, shotgun metagenomics, dietary changes, conservation, *Lepus europaeus*

Amid the ongoing biodiversity crisis, captive breeding and release programs have become vital components of global conservation efforts (1, 2). These initiatives not only provide safe havens for threatened species but also create genetically diverse source populations for reintroduction into their natural habitats (3). Despite big efforts to adjust captivity management practices to each species' needs, the success rate of many reintroduction programs remains limited, with the underlying causes often unclear (4, 5).

Although managers strive to replicate natural habitats as closely as possible, captive-bred animals experience lifestyles that differ significantly from those of their

Address correspondence to Ostaizka Aizpurua, ostaizka.aizpurua@sund.ku.dk.

The authors declare no conflict of interest.

See the funding table on p. 12.

wild counterparts (6, 7). When animals are released into the wild, they face drastic changes in diet, environmental exposure, foraging behavior, and social structures. To mitigate these transitions, many programs incorporate buffering strategies, such as exposure to a wild-like diet or confinement within natural habitats, which need to be tailored to the adaptability of each species (8, 9). Traditionally, the adaptive capacity of animals has been defined as a function of physiological and behavioral traits largely determined by the host genome (10, 11). Recent studies, however, have highlighted the critical role of gut microorganisms in many aspects of animal biology, including nutrition, immune function, and behavior (12–14). In herbivores, the gut microbiota is particularly important because it degrades complex plant fibers, increasing nutrient digestibility (15). In consequence, there is an increasing recognition that gut microbiota may significantly influence the adaptive capacity of animals reintroduced into the wild (16–18).

Research comparing the gut microbiota of captive animals with their wild counterparts has consistently revealed that captivity alters the gut microbiota (19–24). However, the specific responses varied across species (25, 26), indicating that a one-size-fits-all approach to managing animal health through microbiome modulation is unlikely to be effective. Although the exact mechanisms behind these differences are not fully understood, diet is considered a key factor. Due to management constraints, captive animals are often fed simplified dietary formulations, such as chow or pellets (20, 26, 27). Although these diets are nutritionally complete, they lack the structural complexity and indigestible components found in wild diets. These components are crucial in shaping the gut microbiota, particularly in hindgut-fermenting herbivorous animals, as they are necessary for the production of essential fermentation metabolites like short-chain fatty acids, which have a direct impact on host health (28, 29). As a result, maintaining, or gradually transitioning, captive animals to a more natural diet has been proposed as a strategy to enhance their well-being and improve the success of reintroduction into the wild.

One of the reasons why the mechanisms behind the varying microbiota are poorly understood is that most prior studies have relied on 16S rRNA amplicon sequencing (30–32), a method that, while useful, does not provide direct insights into the functional differences between microbial communities in captive versus wild animals. In contrast, genome-resolved metagenomics, which reconstructs bacterial genomes from metagenomic samples, enables the recovery of the functional capacities of each member of the gut microbiota (33). This approach allows for a more detailed understanding of the metabolic contributions microorganisms can offer to their host, thus providing a much deeper resolution on the mechanistic processes behind microbial turnover between captive and wild populations.

In this study, we used the European hare (*Lepus europaeus* Pallas, 1778) as a study system to analyze the functional differences in gut microbiota between wild and captive animals using genome-resolved metagenomics. We leveraged an ongoing captive breeding and reintroduction program in which animals undergo a dietary transition from exclusively pellet-based to a grass-supplemented pellet regimen before release. We tested whether supplementing the diet of captive hares with grass could shift the functional capacities of their microbial communities to more closely resemble those of their wild counterparts, thereby promoting microbial adaptations better suited for digesting their natural diet. By doing so, we aim to deepen our understanding of the role gut microbiota plays in the success of reintroduction programs and inform more effective conservation strategies for the European hare.

## MATERIALS AND METHODS

### Captive-breeding program

This research was conducted within the framework of the captive-breeding program of the Regional Government of Gipuzkoa (Basque Country), located at the southern edge of the European hare's geographical distribution. This action aims to restock

hare populations in the region, which have undergone a significant decline over the past four decades. The program involves a carefully managed reintroduction process, where captive-bred hares are gradually acclimatized to the wild environment. A key part of this process involves supplementing their pellet-based diet (140 g per animal; C1, Unamuno, Altsasu) with 200 g of dry meadow grass (dominated by *Anthoxanthum odoratum, Arrhenatherum elatius, Dactylis glomerata, Trisetum flavescens*, and *Trifolium pratense*) every 2 days, starting 10 days before their release into the natural habitat. Despite these efforts, some animals experience adaptation challenges, such as diarrhea, after being released.

## Sampling

A total of 24 fecal samples were obtained from animals bred at the hare breeding facility in Altsasu, The Basque Country (42.88°, −2.19°). Fecal samples were collected twice from each individual: once before ($n = 12$) and 10 days after grass ($n = 12$) was introduced into their diet. In addition, samples from 21 wild hares were also collected by leveraging animals captured by hunters in the region. Immediately after the animals were hunted, forest rangers from the Regional Government of Gipuzkoa collected intestinal samples directly from the distal colon/rectum to collect the intestinal samples that tend to exhibit the closest resemblance to feces (34, 35). In short, the intestine was transversely opened from the rectum, and intestinal contents were collected using sterile inoculation loops, minimizing contact with the animal tissue and blood. All intestinal and fecal samples were placed in tubes containing 1 mL of DNA/RNA Shield (Zymo, USA) and stored in a refrigerator before being shipped to the lab, where they were kept at −20°C until DNA extraction.

## Laboratory work

Sample processing was conducted following the EHI data generation procedures (https://www.earthhologenome.org/laboratory/) (36, 37). In short, chemically digested samples were subject to mechanical cell lysis (bead beating) using Tissuelyzer (Qiagen, USA) to release as much DNA as possible from different kinds of prokaryotic cells. Subsequently, DNA was isolated and purified using silica magnetic beads combined with solid-phase reversible immobilization to remove as many inhibitors as possible using the DREX1 protocol (38). Extracted DNA was quantified using a Qubit 3 fluorometer (Thermo Fisher Scientific) and the DNA quantities were leveled to 200 ng of DNA in 24 µL of water before shearing the DNA using a Covaris LE220 ultrasonication device to adjust DNA fragment sizes to efficient short-read sequencing. Adapter ligation-based sequencing library preparation was then conducted using the in-house developed BEST protocol (39). Library quality was screened through qPCR and gel electrophoresis before conducting library amplification with dual index identifiers with an adjusted number of PCR cycles for each library to minimize the number of artifacts and loss of library complexity derived from sample overamplification. Finally, uniquely indexed libraries were merged in multiple library pools at exact molarities to produce around 8 Gb (gigabases) of 150 bp paired-end sequencing data per sample. Sequencing was carried out in multiple sequencing runs at an Illumina NovaSeq X platform by the sequencing provider Novogene (UK).

## Bioinformatics

Data processing was carried out using the standard EHI bioinformatics pipeline (https://www.earthhologenome.org/bioinformatics/). The pipeline is based on Snakemake (40) for workflow management, conda for environment management, and SLURM for computational job management (41), and it is directly managed from the EHI database. Data were first quality-filtered using fastp (42) before splitting the metagenomic fraction from the host genomic fraction through mapping reads against the reference *L. europaeus* genome (GCF_033115175.1) using bowtie2 (43). Subsequently, the

metagenomic fraction underwent the genome-resolved metagenomic pipeline. Samples were both assembled individually and co-assembled at the origin level using Megahit (44). Assembly contigs were clustered into bins using Maxbin2 (45), Metabat2 (46), and CONCOCT (47), followed by bin refinement using Metawrap (48), and bin quality assessment using CheckM2 (49). Only bins with completeness values over 50% and contamination values under 10% were considered for downstream analyses (50). All resulting metagenome-assembled genomes (MAGs) were dereplicated using dRep (51), before annotating them taxonomically against the GTDB database using GTDB-tk (52) and functionally against the Pfam, KEGG, UniProt, CAZY, and MEROPS databases using DRAM (53). Finally, quality-filtered reads from each sample were mapped against the annotated MAG catalog using bowtie2 (43) to quantify the representation of each MAG in each sample using CoverM (54). We used *distillR* package (https://github.com/anttonalberdi/distillR) to convert functional annotations into Genome-Inferred Functional Traits (GIFTs). This package provides quantitative measures of each MAG's ability to degrade or produce important biomolecules, using a reference database containing 328 metabolic pathways and modules sourced from the KEGG (55) and MetaCyc (56) databases, which enables the transformation of raw annotations into 190 distinct GIFTs.

## Data analysis

All statistical analyses were conducted using R software version 4.3.2 (R Core Team, 2023). We calculated the alpha diversities of microbial communities using Hill numbers (57). To capture the effects of different diversity components (richness, neutral, phylogenetic, and functional) and diversity orders ($q = 0$ considers only presence/absence, while $q = 1$ gives weight to MAGs based on their relative abundances), we calculated species richness at $q = 0$, neutral diversity at $q = 1$, phylogenetic diversity at $q = 1$, and functional diversity at $q = 1$ using the *Hilldiv2* package version 2.0.2 (58). Alpha diversity differences were determined by a parametric *t*-test when the data were normally distributed (Shapiro-Wilk test) and when the variances of the two groups were equal. When this assumption was not held, a non-parametric Wilcoxon test was performed. Linear mixed models, as implemented in the *lme4:lmer* function (59), were employed to evaluate differences in bacterial alpha diversity between captive-bred individuals before and after the grass was included in the diet, with hare ID included as a random effect.

We calculated the MAG composition dissimilarities between different samples using Hill numbers by computing the Jaccard-type turnover for neutral, phylogenetic, and functional beta diversities at order $q = 1$ using *hilldiv::hillpair*. To visualize the variation in microbial composition, we performed nonmetric multidimensional scaling ordination plots based on the derived distance matrices. Differences in dispersion within sampling methods were assessed using the *betadisper* function in the *vegan* package (60). To test for differences in microbial composition between samples, we conducted a PERMANOVA using the *adonis2* function in *vegan*. When comparing captive-bred individual differences before and after the change in diet, hare ID was included as a blocking factor to control for repeated sampling using the *strata* function. Additionally, to visualize bacteria according to their functional traits, MAGs were ordinated based on their GIFTs through a t-SNE analysis using the *Rtsne* package version 0.17 (61). Metabolic capacity index was calculated as a quantitative metric of the metabolic contributions that gut microbiota can confer to their hosts using *distillR*.

We performed differential abundance analyses to identify microbial taxa that significantly differ between samples using *ANCOM-BC2* (62). These analyses accounted for the random effect of hare ID when analyzing repeated individuals before and after the dietary change. Differential abundance analyses were conducted at the MAG and phylum levels. Additionally, we calculated community-weighted values of GIFTs before comparing values between groups using the Wilcoxon test. The *P*-values were adjusted to account for multiple tests using the Bonferroni method.

## RESULTS

### A genome catalog of the European hare

We generated a total of 222.06 Gbp of sequencing data from 45 samples, with an average depth of 4.93 ± 1.08 Gbp per sample. The co-assembly and binning of the metagenomic data yielded 860 metagenome-assembled genomes (Fig. 1A), with an average completeness of 80.99% ± 16.79% and contamination of 2.21% ± 2.32% (Fig. S1). The MAG catalog was dominated by Bacillota A (61%) genomes, followed by Bacteroidota (14.77%), with the rest of the phyla accounting for less than 10% of the genomes. Notably, 79 MAGs were not annotated at the genus level, and 749 MAGs (87.09%) lacked species-level annotations that differ depending on the phylum, from 33.85% (Bacteroidetes) to 100% (Actinomycetota, Patescibacteria, Desulfobacterota, Bacillota B, and Campylobacterota) (Fig. 1A). Overall, Bacillota A had the highest number of MAGs without species-level annotation, accounting for 516 MAGs (98.1%). MAGs exhibited a genome size range spanning 0.6 Mb (*Nanosyncoccus*, Patescibacteria) and 9.9 Mb (*Lawsonibacter*, Bacteroidota). The dereplicated MAG catalog contained 1,362,707 redundant genes, out of which 1,045,555 (76.7%) received some annotation and 610,762 (44.8%) were assigned KEGG orthologs. MAGs exhibited a wide range of functional attributes, with Bacillota (A, B, and C) showing the widest functional capacities (Fig. 1B). MAGs belonging to the same phylum clustered together, suggesting that MAGs that are phylogenetically closer have higher similar functional attributes.

When the reads were mapped to the host genome, we observed that some wild samples contained a significantly higher fraction of host DNA, which caused an underrepresentation of microbial data. To address this, samples with a high host DNA fraction (*n* = 9) were excluded from community analyses (Fig. S2). This exclusion decreased the sample size to 36 (12 wild, 12 captive pre-grass, and 12 captive post-grass), but increased the average values and decreased the dispersion of domain-adjusted mapping rates, from 86.5% ± 27.77% to 100%, indicating that the sequencing effort

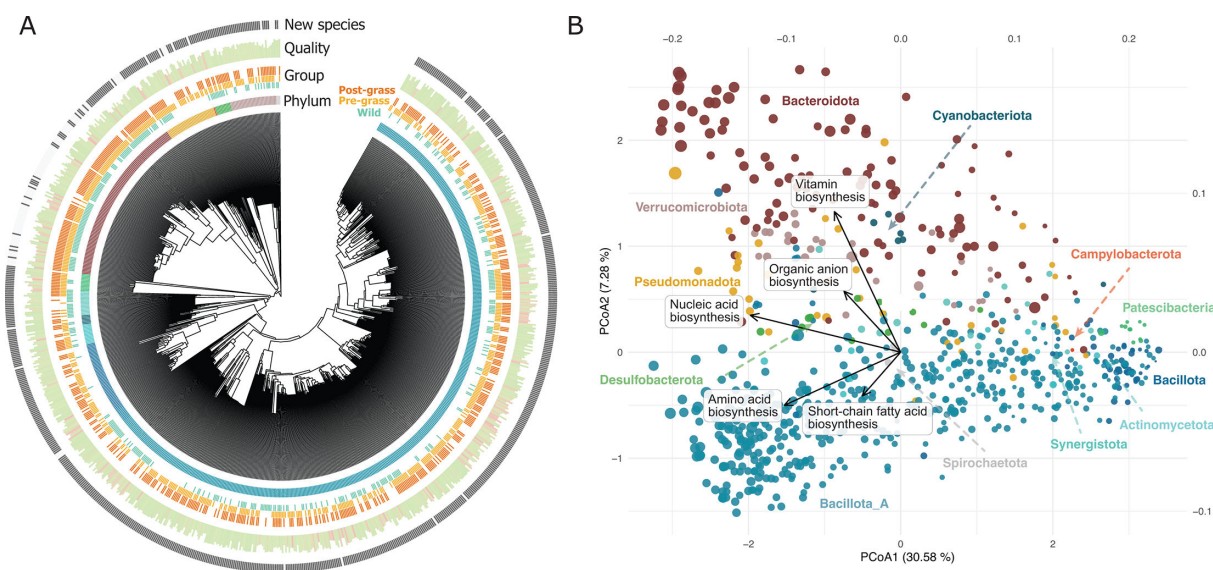

**FIG 1** Overview of the metagenome-assembled genome catalog. (A) Circular phylogenetic tree of the 860 MAGs reconstructed in this study. The first ring displays the phylum of the genome. The second ring indicates the quality of the genome, with the length of the bars showing completeness and color indicating a contamination gradient from 0 (green) to the maximum value of 10 (red). The third ring indicates whether the genome is a new species (dark) or related (>95% average nucleotide identity) to a known species. (B) PcoA ordination of the functional attributes of the genomes, colored by phylum. Each dot represents a bacterial or archaeal genome, with size indicating genome length. The horizontal axis captures a genome size gradient, as indicated by the decreasing dot sizes from left to right. The vertical axis captures a variation in major functional features, as indicated by the eigenvectors of the five metabolic functions with the largest eigenvalues.

carried out was sufficient to surface virtually the entire microbial diversity present in the samples.

## Taxonomic and functional composition of wild and captive hares

The gut microbiota of European hares revealed the presence of 14 bacterial phyla, with Bacillota A (57.24% ± 12.62%) and Bacteroidota (25.79% ± 10.11%) being the most dominant across all samples (Fig. 2). These 14 bacterial phyla were consistently observed in both wild and captive-bred individuals, although significant differences in their relative abundances were noted (Table S1). While Bacillota A and Bacteroidota were the two dominant phyla in both groups, eight phyla were significantly different (Spirochaetota, Bacillota B, Synergistota, Bacillota C, Desulfobacterota, Verrucomicrobiota, Bacteroidota, and Bacillota A, $P < 0.05$) (Fig. 2). For example, wild hares exhibited a notable presence of Spirochaetota (11.73% ± 14.79%), which was nearly absent in captive individuals (0.00063% ± 0.0007%). Additionally, Archaea, specifically Methanobacteriota, were detected exclusively in captive-bred individuals.

Further analysis revealed six MAGs from Pseudomonadota (present in 55.56% ± 16.39% of the samples), one MAG from Bacillota A (50%), and one Archaea MAG from Methanobacteriota (66.67%) to be unique to the captive-bred hares. One MAG was found to be present only in wild individuals (Pseudomonadota), but it only appeared in two individuals. Differential abundance analysis identified 342 MAGs and 30 genera that were significantly different between wild and captive-bred hares (Fig. 3C). Taxonomic diversity analyses revealed significant differences in both alpha and beta diversities between wild and captive-bred hares. Specifically, alpha diversity analyses indicated higher richness and neutral diversity in captive hares (Fig. 3A), though phylogenetic diversity did not show significant variation. All metrics were statistically significant in beta diversity, with neutral diversity ($R^2 = 0.254$, $P = 0.001$, Fig. 3B) and phylogenetic diversity ($R^2 = 0.308$, $P = 0.001$) showing clear distinctions between wild and captive-bred hares.

In terms of gut microbiota functional capacities, functional alpha diversity was found to be statistically significant ($t = 4.3493$, df = 21.934, $P$-value = 0.00026), whereas functional beta diversity was not ($R^2 = 0.188$, $P = 0.086$). The metabolic capacity index values did not differ significantly between the two groups (0.249 ± 0.0152 in captive-bred versus 0.254 ± 0.0206 in wild individuals; $t = -0.65356$, df = 20.27, $P$-value

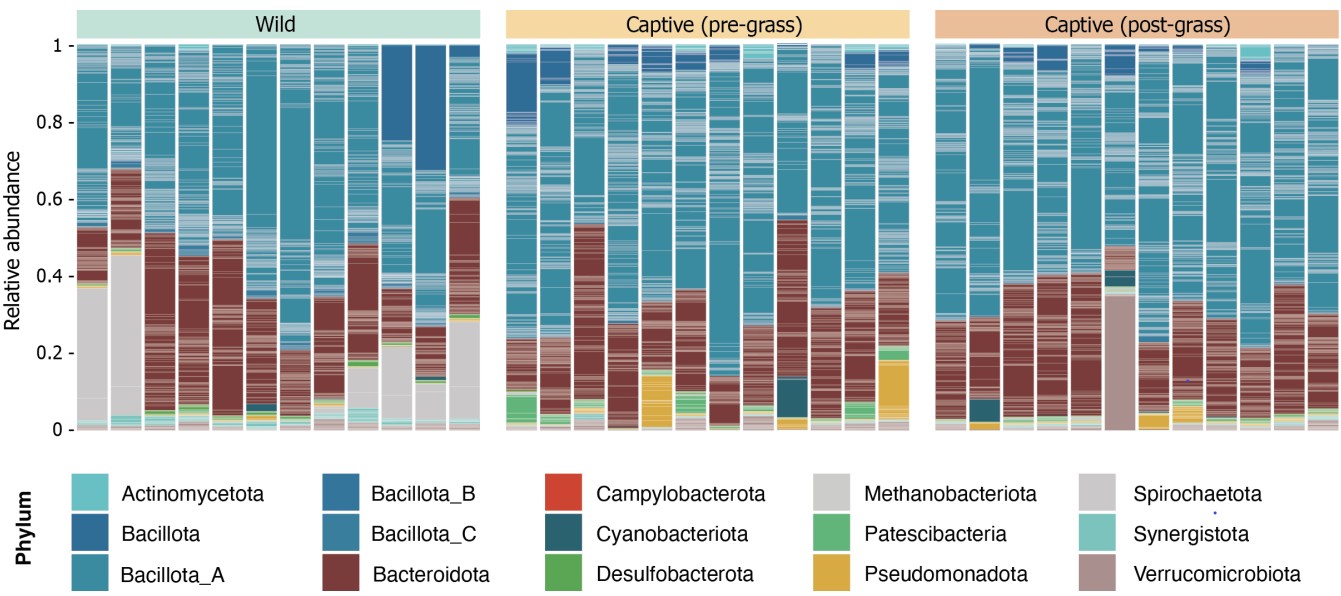

**FIG 2** Community composition of wild and captive hares. Each column represents a sample, while tiles represent the relative abundances of MAGs colored by phylum.

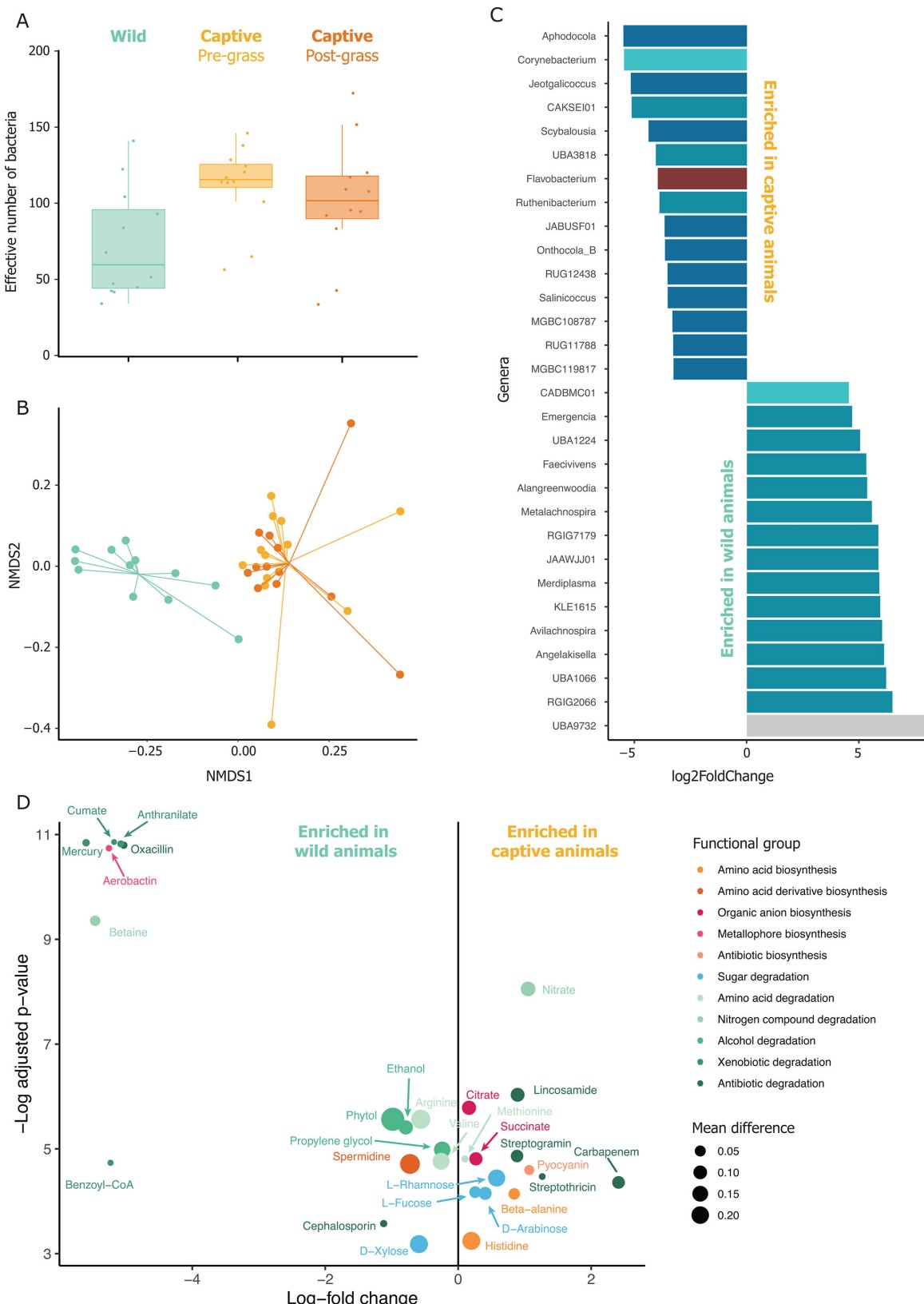

**FIG 3** Diversity, taxonomic, and functional microbiome differences between wild and captive hares. (A) Alpha diversity differences between the three analyzed groups in terms of Hill numbers of $q = 1$ (exponential of Shannon index). (B) Beta diversity differences between the three analyzed groups in terms of Hill numbers of $q = 1$. (C) Differential abundance of genera between wild- and captive-born animals. Genera with significant log-fold differences between sample

**Fig 3 (Continued)**

types are named by genus and colored according to their phyla. (D) Differential representation of microbial community-wide metabolic functions (capacity to biosynthesize or degrade metabolic compounds) between wild- and captive-bred animals. The *x*-axis shows the log-fold difference in community-weighted functional trait values, while the *y*-axis indicates the adjusted *P*-value of these differences. Dot size represents the absolute difference in function completeness, reflecting the effect size of the observed changes.

= 0.521). However, 14 functions were enriched in captive individuals, while 15 were enriched in wild individuals (Fig. 3D). In captive hares, pathways for four antibiotics, an amino acid, a nitrogen compound, and two sugar degradations were enriched, as well as the biosynthesis of two organic anions and two amino acids, and an antibiotic pyocyanin. On the contrary, wild hares possessed gut bacteria with a higher capacity for degrading amino acids, four xenobiotics, three alcohols, two antibiotics, nitrogen compounds, D-xylose sugar, as well as synthesizing the amino acid derivative spermidine and the metallophore aerobactin compared to their captive-bred counterparts.

## Effect of dietary switch to grass in captive hares' microbiota

To determine if the introduction of grass into the diet could pre-adapt the gut microbiota of European hares and enhance their adaptation capacity, we first assessed whether this dietary change produced any alterations in microbial composition and function in captive-bred animals. Our analysis revealed no significant differences in alpha (Fig. S3) and beta (Fig. S4) diversity in the individuals before and after grass was included in the diet ($P > 0.05$). However, differential abundance analysis identified 11 MAGs that differed between the groups (Fig. S5). Of these, eight MAGs were enriched in hares on a diet without grass, while three MAGs increased in abundance following the inclusion of grass in the diet. Specifically, the introduction of grass resulted in the increased presence of two MAGs from the Lachnospiraceae and Oscillospiraceae families within the Bacillota A phylum, and one MAG from the Xanthomonadaceae family within the Pseudomonadota phylum. In terms of functional differences, only two traits showed significant variation between the groups. The abundance of bacteria with the capacity to biosynthesize the metallophore staphylopine and degrade the antibiotic chloramphenicol decreased when grass was added to the diet.

## DISCUSSION

Our metagenomic analyses of wild and captive European hares (*L. europaeus*) represent the first effort to reconstruct the species' gut microbiota using genome-resolved metagenomics. Beyond characterizing microbial diversity and composition, this approach enabled us to assess the functional potential of the microbiome, offering insights into its role in shaping host biology. Although the scale of the captive breeding program and the challenges of collecting intestinal samples from wild animals limited the sample size to 45, our in-depth microbiome analysis revealed clear differences between captive and wild hares, as well as insights into the effects of transitioning from an exclusively pellet-based to a grass-supplemented diet before release.

### The diverse microbiome of European hares

The metagenomic assembly and binning of the fecal samples from 45 wild and captive hares yielded 860 high-quality genomes, spanning 14 bacterial and 1 archaeal phyla. This broad taxonomic representation is comparable to other hindgut-fermenting herbivores and reflects the vast number of ecological niches herbivorous diets generate in the lower intestine (63). In accordance, microorganisms exhibited a wide range of functional capacities, distributed across a complex landscape of metabolic potential. In line with previous studies based on 16S rRNA gene sequencing (32), the gut microbiota of hares was highly dominated by members of Bacillota A and Bacteroidota, taxa known for their role in breaking down complex polysaccharides from plant fibers and producing short-chain fatty acids as a result of their fermentation.

Noteworthily, the vast majority (87%) of reconstructed genomes exhibited average nucleotide identities below 95% from any previously recorded reference genomes, which is in line with other wildlife microbiome analyses that also reported novel species rates over 75% (36, 64). These observations indicate knowledge biases toward model species, with wild organisms still serving as reservoirs of vast and largely uncharacterized microbial biodiversity. Similarly, 23.3% of predicted genes in our data set lacked functional annotation, indicating that a significant portion of microbial functional potential remains unknown (65). This high biodiscovery rate reflects the immense, largely unexplored diversity within vertebrate-associated microbial communities and highlights the limitations of relying solely on 16S rRNA sequencing and current reference databases for functional inference in wild species. Our findings emphasize the value of genome-resolved shotgun metagenomics not only for detecting novel taxa but also for uncovering microbial functions directly from the genomes reconstructed from the system under study, rather than relying on indirect inference from genomes characterized in other environments, which can often be functionally distinct (66). This approach offers critical insight into host-microbiota interactions and substantially enhances our understanding of microbial ecology in wildlife systems.

## Differences between captive and wild hares

Our analyses unveiled significant differences in the taxonomic composition and functional capabilities of the gut microbiota between captive and wild European hares, a trend that has been observed across various vertebrate species (22, 25, 26, 67–69). The transition to or breeding within a captive environment usually leads to shifts in microbial communities, which appear to be largely influenced by species-specific and environmental conditions within captivity (25). Although captivity-induced shifts in the microbiota are common, the underlying processes remain unclear, and it is still uncertain whether these changes result from dysbiosis, reflect adaptive responses to new environmental conditions, or constitute neutral changes with limited consequences for host biology.

Age differences between groups could, in principle, contribute to the observed patterns, as captive individuals were on average 5 months old, whereas wild hares were over 1 year old. However, longitudinal and cross-sectional studies in lagomorphs and other mammals indicate that the core gut microbiota typically reaches an adult-like composition shortly after weaning (70–72). Subsequent ontogenetic changes mainly influence the relative abundances of existing taxa rather than the presence or absence of entire phyla. In this context, the combination of (i) identical phylum-level composition in captive and wild hares—both groups shared the same 14 bacterial phyla, with Bacillota and Bacteroidota dominating, consistent with previous findings in lagomorphs (73–75)—and (ii) higher diversity in the younger, captive group argues against age as the primary explanation for the differences detected. Instead, these patterns point to captivity-related factors as the main drivers introducing additional microbial lineages and altering their relative abundances as observed in rabbits (76). Among these, seven bacterial species were unique to captive hares, comprising six Pseudomonadota (five MAGs from Gammaproteobacteria and one from Alphaproteobacteria) and one Bacillota A (Clostridia). Additionally, the only archaeal genome reconstructed, *Methanosphaera cuniculi*, was also exclusive to captive hares. *Methanosphaera* has been reported in a range of animals, such as squirrels (77), rabbits (78), pigs (79, 80), humans (81), and ruminants (82), highlighting its adaptability to different host gut environments. Notably, *Methanosphaera* exhibits host specificity, with reduced genome sizes in monogastric hosts (~1.7 Mbp) compared to larger genomes found in ruminants (~2.9 Mbp) (83). The genome size of the reconstructed *Methanosphaera* in captive hares (1.7084 Mbp) closely resembled those found in monogastric animals (83). Although these archaea are well-established contributors to fiber fermentation in ruminants and other monogastric herbivores, their role in hares remains less defined. These archaea can contribute to hydrogen turnover in the gut, thereby potentially aiding fermentation efficiency. Yet, in our study, *Methanosphaera* was not present in wild hares, and its absence did not

appear to affect polysaccharide degradation capacities, suggesting that it may not be essential for fiber fermentation in hares. This finding contrasts with studies in ruminants, where methanogens play a more central role in fermentation, converting hydrogen into methane to maintain balance in the gut environment (84). The lack of methanogens in wild hares, alongside comparable microbial degradation capacities between wild and captive animals, supports the view that archaea may not be necessary for optimal fermentation in monogastric herbivores like hares in natural settings. However, their potential significance in captivity remains an open question for further investigation.

The relative abundances of several bacteria differed significantly between captive and wild hares, with the most notable difference observed in the phylum Spirochaetota. This phylum has previously been reported as a prominent component of hare microbiota, with a larger representation compared to rabbits (73). The two MAGs identified in this study lacked species-level annotation but were classified within the genus *UBA9732* (Sphaerochaetaceae). Despite being the third most abundant phylum in wild hares, Spirochaetota were nearly absent in captive-bred individuals. This absence could be attributed to several factors, including the unavailability of these bacteria in captivity or the younger age of the captive hares, as Spirochaetota may colonize hare guts later in life. However, a prior study also reported no Spirochaetota among the top genera in captive hares (32), reinforcing the idea that limited exposure to these bacteria in captivity may be a contributing factor.

Interestingly, *UBA9732* has also been found in the rumen of herbivores (85) and in the gut microbiomes of populations with high fiber intake, such as South Africans (86) and Baka forager-horticulturalists from Cameroon (87). The enrichment of Spirochaetota has been observed in rural populations (88, 89), while a significant decline in these bacteria has been noted in westernized humans and platyrrhines, likely due to lifestyle changes (90). Our findings suggest that the limited presence of these anaerobic, non-spore-forming bacteria in captive hares is likely a result of altered diets and reduced social interaction in captivity (90). The two Spirochaetota genomes reconstructed in this study exhibited strong capacities for degrading alpha- and beta-galactans, as well as arabinan, which could enhance the digestion of complex plant fibers in the lower intestines of hares. This ability may be particularly advantageous for wild hares consuming fiber-rich diets, but less relevant for captive hares on controlled, processed diets (91).

Overall, the gut microbial communities of wild and captive hares displayed notable differences in the functional capacities directly inferred from their reconstructed genomes. Wild hares exhibited enhanced microbial abilities to degrade D-xylose, while captive-bred hares showed enriched capacities for degrading L-rhamnose and L-fucose. D-xylose is abundant in hemicellulose, a major component of plant cell walls, particularly in hardwoods and grasses. In contrast, L-rhamnose and L-fucose are abundant in softer plants. Although without gene expression and metabolomic analyses, we cannot ascertain whether these functional capacities are actually expressed, the distinct sugar degradation profiles observed between wild and captive hares probably reflect their adaptation to diets with distinct sources of carbohydrates.

Similarly, wild hares were found to host gut microbiota with greater capacities for amino acid degradation compared to their captive counterparts. This enhanced microbial function may also be linked to the dietary preferences of hares, particularly during autumn and winter, when they tend to favor plants that are low in fiber but rich in crude fat and protein (92). These seasonal dietary shifts could select gut microorganisms with an increased ability to break down amino acids, providing hares with essential nutrients during periods when high-protein foods are more abundant. This functional adaptation in their microbiota likely helps wild hares maximize nutrient extraction from their protein-rich diet, supporting their survival in changing environmental conditions. In contrast, amino acid biosynthesis was found to be higher in captive hares, suggesting that their gut microbiota might compensate for a diet potentially lower in protein by producing essential amino acids. This increased biosynthetic capacity may reflect a more controlled and less varied diet in captivity, where external protein sources are limited.

Furthermore, the gut microbial community of wild hares exhibited a higher capacity to degrade xenobiotics, although the mean difference between wild and captive hares was relatively small. We speculate that the presence of bacteria with the ability to break down xenobiotics might be because of the exposure of wild hares to these environmental contaminants (93) or, indirectly, it turns out that the necessary bacteria found in wild hares have these capabilities as well. In natural environments, animals encounter various xenobiotics, including pollutants and plant- or industry-derived chemicals. Future studies could explore whether this microbial capacity for metabolizing xenobiotics functions as a protective adaptation (94), helping wild hares detoxify and process harmful substances like mercury, which pose serious risks to animal health (95).

## Effect of dietary switch to grass in captive hares

Gut microbiota has been proposed as a tool to pre-adapt captive-bred animals to wild conditions, potentially improving the success of reintroduction programs (27, 31). In this study, we examined whether switching captive-bred hares from a pellet-based diet to grass could modify their gut microbial community, aligning it more closely with that of wild hares. Contrary to a previous study (32), we observed that this dietary change led to only minimal alterations in the gut microbiota. For instance, we notice a decrease in *Eggerthellaceae* after the introduction of grass. This bacterial family is associated with a chow diet in mice, likely due to its high content of cereal-derived fibers (96). Following the dietary switch, we also observed an increase in the relative abundance of *Luteimonas*. This genus has been detected in the phyllosphere of various plants (97), suggesting that hares may have been exposed to this bacterium through their new food source. Similarly, the microbial community showed few functional changes. The decrease in bacteria that produce staphylopine—a metallophore associated with bacterial virulence—after grass introduction could potentially reduce the infection risk of the animals and improve animals' health in captivity. Additionally, the observed loss of the capacity to degrade chloramphenicol may be linked to the decrease in *Eggerthellaceae*, a bacterial family known for its high capacity to metabolize this compound. While the relative abundances of certain bacteria differed, the inclusion of grass in the captive diet did not include new bacterial species or significantly enhance the functional capacities of hares. Crucially, the 10-day dietary intervention failed to shift the microbial community of captive hares toward that of their wild counterparts in a way that would pre-adapt them to the natural environment. This limited response may reflect the short duration of the intervention, as microbial community rearrangements may require a more extended period to yield significant differences (98).

Alternatively, the differences in fiber and amino acid degradation capacities between captive and wild hares suggest that using specific plant species consumed by wild hares —particularly those rich in protein and complex carbohydrates—rather than generic grass, might be crucial in aligning captive microbiomes with those of wild hares. A detailed analysis of wild hares' dietary habits could help identify these critical plant sources. Incorporating them into the diet of captive hares could serve as a more targeted and effective strategy for microbiome pre-adaptation, ultimately enhancing the success of reintroduction efforts.

## Conclusion

This study demonstrated the value of moving beyond the widely used 16S rRNA sequencing approach by integrating genome-resolved metagenomics with functional microbiome analyses to uncover key differences in the gut microbial communities of wild and captive European hares. Our findings reveal that adding grass to the captive diet in this reintroduction program does not sufficiently align captive and wild microbiomes. The functional discrepancies observed suggest that successful animal pre-adaptation strategies require a more nuanced understanding of species-specific dietary habits, rather than relying on superficial dietary modifications. To improve reintroduction outcomes, it will be essential to develop diets that not only mimic natural plant

compositions but also support critical microbial functions necessary for adaptation, such as preventing diarrhea. In cases where dietary intervention proves logistically challenging, alternative strategies such as "natural training" may facilitate microbiome adaptation more effectively (91). This strategy relies on releasing animals into enclosed environments containing the full diversity of their native dietary sources, which could help animals transition more smoothly to wild conditions, reducing health risks that might otherwise compromise survival.

Future multi-omic analyses will be instrumental in clarifying how microbial functional capacities are expressed and regulated, offering deeper insights into the diet–microbiota–host axis. Such integrative approaches will help unravel the mechanisms by which specific microbial functions contribute to host adaptation, health, and resilience. In parallel, longitudinal studies tracking the temporal dynamics of microbiome shifts following environmental or dietary changes will be necessary for understanding how long it takes animals to reconfigure their gut microbiota in response to natural conditions. This knowledge will contribute to improved designing of effective dietary interventions that support microbiome pre-adaptation prior to reintroduction. As habitats continue to change and human impacts on ecosystems intensify, understanding how gut microbiota influences animal adaptation and health will be increasingly vital for the success of species rewilding and conservation efforts.

## ACKNOWLEDGMENTS

We thank all the hunters and forest rangers from the Regional Government of Gipuzkoa, and Leocadio Galán, who participated in the fieldwork and provided the samples, as well as the support people who facilitated the administrative paperwork, with a special mention to Iñaki Olano, who coordinated the sample collection. Finally, we thank Prof. Tom Gilbert for supporting the study and for his valuable feedback on the manuscript.

This work was supported by the Carlsberg Foundation through the grant CF20-0460 and the Danish National Research Foundation under the grant DNRF143 "A Center for Evolutionary Hologenomics." The funders had no role in study design, data collection and interpretation, or the decision to submit the work for publication.

## AUTHOR AFFILIATION

[1]Center for Evolutionary Hologenomics, Globe Institute, University of Copenhagen, Copenhagen, Denmark

## AUTHOR ORCIDs

Ostaizka Aizpurua  http://orcid.org/0000-0001-8053-3672
Garazi Martin-Bideguren  http://orcid.org/0000-0003-0157-9125
Antton Alberdi  https://orcid.org/0000-0002-2875-6446

## FUNDING

| Funder | Grant(s) | Author(s) |
| --- | --- | --- |
| Carlsbergfondet | CF20-0460 | Antton Alberdi |
| Danmarks Grundforskningsfond | DNRF143 | Antton Alberdi |

## AUTHOR CONTRIBUTIONS

Ostaizka Aizpurua, Conceptualization, Data curation, Formal analysis, Methodology, Project administration, Writing – original draft, Writing – review and editing | Garazi Martin-Bideguren, Conceptualization, Data curation, Writing – review and editing | Nanna Gaun, Data curation, Methodology, Writing – review and editing | Antton Alberdi,

Conceptualization, Formal analysis, Funding acquisition, Methodology, Visualization, Writing – original draft, Writing – review and editing

## DATA AVAILABILITY

Raw sequencing data and microbial genome sequences were published in the 1st EHI Data Release (99) under Bioproject PRJEB76898. The data tables containing the quantitative information analyzed and the R scripts used for statistical analyses are available in a dedicated GitHub repository (https://github.com/alberdilab/lepus_metagenomics), both rendered into a HTML web-book (https://alberdilab.github.io/lepus_metagenomics) and archived in Zenodo with doi: 10.5281/zenodo.18780230.

## ADDITIONAL FILES

The following material is available online.

### Supplemental Material

**Supplemental table and figures (Spectrum03691-25-s0001.docx).** Table S1 and Figures S1 to S5.

### Open Peer Review

**PEER REVIEW HISTORY (review-history.pdf).** An accounting of the reviewer comments and feedback.

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
