## [Reviewer comments · Microbiology Spectrum]

Microbiology Spectrum

Grass supplementation to pellet-based diet fails to enrich gut microbiomes with wild-like functions in captive-bred hares

Ostaizka Aizpurua, Garazi Martin-Bideguren, Nanna Gaun, and Antton Alberdi

Corresponding Author(s): Ostaizka Aizpurua, Kobenhavns Universitet

Review Timeline:

Submission Date:	January 7, 2026
Editorial Decision:	January 19, 2026
Revision Received:	January 23, 2026
Accepted:	January 28, 2026

Editor: Jan Claesen

Reviewer(s): The reviewers have opted to remain anonymous.

Transaction Report:

DOI: <https://doi.org/10.1128/spectrum.03691-25>

Re: Spectrum03691-25 (**Gross supplementation to pellet-based diet fails to enrich gut microbiomes with wild-like functions in captive-bred hares**)

Dear Dr. Ostaizka Aizpurua:

Thank you for the privilege of reviewing your work. Below you will find my comments, instructions from the Spectrum editorial office, and the reviewer comments.

Thank you for carefully addressing the Reviewers' comments from your prior submission to mSystems. I am pleased to inform you that your manuscript has been editorially accepted for publication. However, there are a few additional questions in the Spectrum-specific submission form that need to be answered before the final decision. Once these are completed, please return your submission so that I can move your paper forward to acceptance.

Revision Guidelines

Sincerely,
Jan Claesen
Editor
Microbiology Spectrum

Re: Spectrum03691-25R1 (**Grass supplementation to pellet-based diet fails to enrich gut microbiomes with wild-like functions in captive-bred hares**)

Dear Dr. Ostaizka Aizpurua:

Thanks for addressing the Spectrum-specific items in the submission system. Your paper has now been accepted for publication, congratulations!

Your manuscript has been accepted, and I am forwarding it to the ASM production staff for publication. Your paper will first be checked to make sure all elements meet the technical requirements. ASM staff will contact you if anything needs to be revised before copyediting and production can begin. Otherwise, you will be notified when your proofs are ready to be viewed.

Sincerely,
Jan Claesen
Editor
Microbiology Spectrum